# Autophagy in Rheumatic Diseases: Role in the Pathogenesis and Therapeutic Approaches

**DOI:** 10.3390/cells11081359

**Published:** 2022-04-15

**Authors:** Alessandra Ida Celia, Serena Colafrancesco, Cristiana Barbati, Cristiano Alessandri, Fabrizio Conti

**Affiliations:** Rheumatology Unit, Department of Clinical Internal, Anaesthesiolagical and Cardiovascular Sciences, Sapienza University of Rome, 00161 Rome, Italy; alessandraida.celia@uniroma1.it (A.I.C.); serena.colafrancesco@uniroma1.it (S.C.); cristiana.barbati@uniroma1.it (C.B.); fabrizio.conti@uniroma1.it (F.C.)

**Keywords:** autophagy, apoptosis, rheumatic diseases

## Abstract

Autophagy is a lysosomal pathway for the degradation of damaged proteins and intracellular components that promotes cell survival under specific conditions. Apoptosis is, in contrast, a critical programmed cell death mechanism, and the relationship between these two processes influences cell fate. Recent evidence suggests that autophagy and apoptosis are involved in the self-tolerance promotion and in the regulatory mechanisms contributing to disease susceptibility and immune regulation in rheumatic diseases. The aim of this review is to discuss how the balance between autophagy and apoptosis may be dysregulated in multiple rheumatic diseases and to dissect the role of autophagy in the pathogenesis of rheumatoid arthritis, systemic lupus erythematosus, and Sjögren’s syndrome. Furthermore, to discuss the potential capacity of currently used disease-modifying antirheumatic drugs (DMARDs) to target and modulate autophagic processes.

## 1. Introduction

Autophagy is an essential lysosomal pathway for the degradation of long-lived, damaged proteins or intracellular components, aimed at recycling macromolecules in order to maintain cellular metabolic homeostasis. Autophagy can be classified into three main types: macroautophagy, microautophagy, and chaperone-mediated autophagy. Macroautophagy is the process by which cells encapsulate and disrupt defective organelles, while microautophagy refers to the encapsulation of proteins and small organelles directly into lysosomes [1]. Chaperone-mediated autophagy requires the binding of selected proteins to specific heat shock 70 protein (HSPA8), acting as a molecular chaperone that translocates unfolded proteins to lysosomes for degradation [1]. Under physiological conditions, different autophagic processes are crucial for the maintenance of the internal environment, cell homeostasis, and cell survival. It is well known how autophagy increases in response to different extracellular and intracellular stress, such as hypoxia, nutrient deprivation, and infections [2]. Macroautophagy (hereafter simply labeled as “autophagy”) is the best-characterized type of autophagy and plays an essential role in inflammation and immunomodulation. Indeed, several immunological processes are highly dependent on cellular autophagy, including pathogen recognition, antigen presentation, and lymphocyte survival [3,4,5].

While autophagy generally promotes cell survival, apoptosis is a fundamental programmed cell death mechanism. The balance between these two processes is key in determining cell destiny [6]. Several studies have demonstrated that suppression of autophagy, by knocking down autophagy-related genes, results in increased cell death, thus indicating the prominent role of autophagy as a “cell protective” mechanism [6]. This prosurvival function is driven by different mechanisms and exerts its role through the activation of different cell protective pathways designed to eliminate cell stressors such as reactive oxygen species (ROS) and damaged DNA.

Mammalian target of rapamycin (mTOR) is one of the major intracellular autophagy inhibitors [6]. Specifically, when cell growth and protein synthesis are insufficient, mTOR is inactivated with consequent induction of the autophagy pathway. This inhibitory effect of mTOR is implemented by its capacity to phosphorylate Atg13 with subsequent inhibited association with Atg1 (Ulk1/2), an essential upstream molecule in autophagosome cascade formation [6].

Atg5 is a protein critical for the completion of autophagosome formation, although a role in apoptotic processes has been described too. Regarding its role in autophagy activation, after conjugation with Atg12, Atg5 binds to Atg16L and forms a large multiprotein complex (first conjugation system) eventually recruited to the forming autophagosome’s isolation membrane. In the second conjugation system, Atg7 and Atg3 mediate the conjugation of Atg8 (LC3) to the lipid phosphatidylethanolamine resulting in the conversion from its soluble cytoplasmic form (LC3-I) to the membrane-bound autophagosome-associated form (LC3-II), which is required for membrane expansion [6] Figure 1.

However, as mentioned above, it is interesting to note that Atg5 also has a role in apoptotic signaling. Specifically, after cleavage, Atg5 forms an N-terminal product that translocates to the mitochondria and promotes cytochrome C release and caspase activation [6]. Thanks to this capacity, Atg5 is considered a link between the autophagy pathway and the apoptotic pathway [6].

Thus, autophagy appears as a crucial cell survival mechanism that guarantees cell protection and, consequently, a correct balance in cell homeostatic processes. This capacity looks particularly interesting in different cell types, including the immune cell system. Indeed, as demonstrated by previous studies, the balance between autophagy and apoptosis plays a fundamental role in the pathogenesis and progression of several autoimmune diseases [5].

In this review, we discuss how the crosstalk between autophagy and apoptosis can be dysregulated in autoimmune diseases with evidence of aberrant expression of an autophagic process that contributes to the pathogenesis of different rheumatic conditions, including rheumatoid arthritis (RA), systemic lupus erythematosus (SLE), and Sjögren’s syndrome (SS). Current and future therapeutic strategies for these autoimmune conditions will be also considered with particular attention to their capacity of targeting autophagy as part of their mechanism of action.

## 2. Rheumatoid Arthritis

RA is a systemic autoimmune disease characterized by persistent synovitis, systemic inflammation, and autoantibodies production, such as rheumatoid factor and anticitrullinated peptides (ACPAs). ACPAs may be directly pathogenic thanks to their ability to promote synovitis, cartilage disruption, and bone loss via macrophage activation [7]. The contribution of autophagy to the presentation of citrullinated peptides and the generation of ACPAs is a critical step in RA. Several data suggest that increased autophagy leads to the production of citrullinated proteins in RA fibroblast-like synoviocytes (FLS) and that the level of LC3II in FLS positively correlates with the level of ACPAs [8]. As we know, protein citrullination is a post-translational modification catalyzed by the arginine deiminase-4 (PAD-4). It has been shown how, following treatment with a potent autophagy inducer, rapamycin, human synoviocytes exhibit activation of PAD-4 with consequent generation of citrullinated proteins [8]. The additional evidence that PAD-4 is detectable in LC3-II immunoprecipitates from FLS further supports the view that citrullination may occur in autophagosomes [8].

Besides evidence on the role of autophagy in RA citrullination processes, alterations in the balance between autophagic and apoptotic mechanisms in FLS have also been advocated as pathogenic in RA. Specifically, activation of the autophagy pathway appears as a crucial potential mechanism by which RA FLS protect themselves from apoptosis. Of note, FLS apoptosis represents a fundamental mechanism contributing to the reduction in inflammation by blockage of excessive immune cell activation and cytokine production [9]. In this regard, a significant reduction in FLS apoptotic rate and apoptotic mediators in the synovia of patients with RA has been observed [10]; this finding supports the hypothesis of a detrimental role of autophagy in the prevention of self-regulatory anti-inflammatory mechanisms. Increased autophagy-mediated FLS survival looks much more remarkable if we also recall how in RA, these cell types undergo critical molecular changes that lead to a more aggressive and invasive phenotype [11]. Indeed, synovial hyperplasia, FLS infiltration of cartilage, and subchondral bone erosion are hallmarks of RA. Taken together, these findings support the idea that progressive bone and cartilage destruction can be attributable to the resistance of FLS to apoptosis, which is likely mediated by autophagic processes [12]. In line with this hypothesis, previous studies have demonstrated that in RA, FLS undergo hyperplasia as a result of reduced apoptosis, which is mainly indicated by increased levels of antiapoptotic factors and downregulation of proapoptotic factors [13,14,15]. In addition, inhibition of proteasomal activity in FLS seems to increase the intracellular levels of LC3II with consequent permission for increased survival [16]. Immune-histochemical and molecular analysis of autophagy-related molecules in RA synovial biopsies further support these findings, with evidence of increased levels of Beclin1, Atg5, and LC3-II compared to osteoarthritis (OA) [17]. All these data suggest that upregulated autophagy leads to the activation, development, and proliferation of FLS, with consequent promotion of RA-associated synovitis.

In RA patients, FLS do not represent the only cell type possibly affected by maladaptive activation of autophagy. As we know, RA is characterized by subchondral bone erosions promoted by reduced osteoblast-mediated bone formation and increased osteoclast-mediated bone reabsorption. Recent data suggest a possible link between osteoclastogenesis and the autophagy pathway. Specifically, increased expression of autophagy-related molecules, such as Beclin1 and Atg7, in osteoclasts from RA synovia has been observed [18]. Furthermore, in experimental mouse models of arthritis, a significant reduction in bone erosion has been demonstrated following treatment with autophagy inhibitors [18].

## 3. Systemic Lupus Erythematosus

SLE is a complex autoimmune disease with a strong genetic component. Genome-wide association studies have found that single-nucleotide polymorphisms (SNPs) in several autophagy-related genes (ATG5, ATG7, IRGM, DRAM1, CDKN1B, APOL1, and MTMR3) are associated with SLE susceptibility [19]. Specifically, at least five SNPs near the Atg5 locus seem associated with SLE initiation and development [19]. A different study also demonstrated that two specific SNPs in the ATG5 gene (rs6568431 and rs2245214, respectively) are associated, on the one hand, with anemia and renal involvement and, on the other hand, with a higher risk of producing anti-DNA autoantibodies. This finding further suggests that SNPs in autophagy-related genes likely have a role in SLE pathogenesis, determining not only disease susceptibility but also clinical phenotype [20].

Besides genetic evidence, a role for autophagy in SLE pathogenesis has also been detected in different immune-mediated processes relevant to disease pathogenesis, including the removal of dead cells by monocyte and the scavenging of intracellular DNA and RNA, regulation of type I interferon (IFN) responses, and B- and T-cell survival [19]. In SLE patients, difficulty in the clearance of immune complexes and dead cells serves as a perpetual source of autoantigen and leads to an autoimmune response against DNA and RNA products [21,22]. In fact, immune complexes containing autoantibodies against DNA or RNA activate Toll-like receptors (TLRs) 7 and 9, leading to the production of IFN-α by plasmacytoid dendritic cells. Accordingly, autophagy-deficient dendritic cells have decreased TLR7 and TLR9 activation and reduced IFN-α production, but precise molecular mechanisms have not been reported [19,23].

Autophagy is reported to be crucial for monocyte differentiation and for the prevention of regular apoptosis and survival of monocytes. Inhibition of induced autophagy leads to apoptosis [24]. Of note, autophagy shows abnormalities in lupus macrophages, and autophagy-related genes are found to be upregulated in macrophages of lupus mice and SLE patients, suggesting that autophagy may be involved in SLE pathogenesis by influencing monocytes and macrophages [25]. Macrophages from patients with SLE exhibit increased levels of autophagy, and in mice with a lupus-like disease, inhibition of macrophage-induced autophagy leads to decreased B-cell maturation and reduced production of dsDNA [25]. Moreover, adoptive transfer of Beclin1 knockdown macrophages can significantly decrease anti-dsDNA antibody levels and immune complex deposition mitigating proteinuria and glomerulonephritis [25]. This protective effect seems to be associated with the significantly decreased production of IL-6 and TNF-α, indicating that abnormally activated autophagy in macrophages may contribute to lupus by promoting the production of TNF-α and IL-6 [25].

Emerging evidence demonstrates that autophagy is upregulated in SLE B cells during plasma cell differentiation [26]. B-cell-activating factor (BAFF) is one of the mean chemokines involved in SLE pathogenesis driving autoantibody production and is associated with an increased risk of SLE flare [27]. Interestingly, in SLE, BAFF seems to contribute to a dysregulation of B-cell autophagy [28,29]. Specifically, BAFF binding to its receptors (BAFF-R, TACI, and BCMA) results in activation of the noncanonical NFκB signaling pathway and the JNK1 pathway, which promote B-cell maturation and survival through concomitant activation of the autophagy pathway [29]. BAFF signaling through TACI and BCMA can also activate the inhibitor of NF-κB kinase, which in turn modulates downstream autophagy; this finding further suggests how autophagy pathways, promoted by BAFF, might be important for B-cell differentiation and survival in SLE [30].

B cells are not only relevant to SLE pathogenesis, as overactivation of T cells with increased cytokine and autoantigen-mediated signaling has been demonstrated. Interestingly, increased autophagic vesicle formation sustained by the evidence of increased production of LC3II and increased autophagic flux was identified in CD4+ T cells from patients with SLE compared to CD4+ T cells from healthy donors [31].

## 4. Sjögren’s Syndrome

Evidence on the role of autophagy in SS is still limited, although recent studies have demonstrated a potential pathogenic role in this condition too. Previous works from our group demonstrated how dysregulation of the autophagic process is detectable in T and B lymphocytes infiltrating SS minor salivary glands. Specifically, autophagic dysregulation has been detected in both infiltrating and circulating T lymphocytes [32]. The activation of autophagy resulted to be particularly relevant in CD4+ T, with an interesting association between its level of expression and disease histological severity; additionally, the level of autophagy in peripheral blood T lymphocytes positively correlated with patient disease activity and damage indexes [31]. By means of ultraselective methods allowing extracting lymphocyte infiltrates from minor salivary glands, we then demonstrated how upregulation of autophagy is not a mere feature of infiltrating T cells. Specifically, concomitant upregulation of autophagy is detectable in B-cell-infiltrating minor salivary glands with interesting evidence of aberrant activation and a possible pathogenic effect in GC-like structures [33].

Furthermore, it is well known that SS is also considered an “autoimmune epithelitis”, as salivary gland epithelial cells (SGECs) actively participate in the inflammatory process through alteration in homeostatic mechanisms and expression of adhesion molecules [34]. In previous years, different studies have focused their attention on a possible pathogenic role played by autophagy in SS SGECs. For instance, in mice models of dry eye syndrome, autophagy appeared as a crucial survival mechanism aimed at protecting epithelial cells from death [35]. Accordingly, in patients with SS, increased levels of autophagy markers were detected both in tear film and in conjunctiva epithelial cells [36,37]. Evidence of a homeostatic role for autophagy in SGECs also comes from other studies and strongly suggests a role of this mechanism in restoring salivary gland function after external damage [38,39]. However, it is interesting to note that in SS, these attempts to restore salivary gland function via autophagy activation seem to be detrimental. In a very recent work from our group, we demonstrated for the first time the existence of maladaptive activation of autophagy in SS SGECs [40]. Specifically, in SGECs, excessive activation of autophagy sustains cell survival by prevailing upon apoptosis is associated with disease histological severity [40]. This maladaptive activation of autophagy is induced by inflammation and, surprisingly enough, is a driver of SGECs activation [40]. Thus, autophagy appears as a mechanism that allows SGECs to “struggle for survival” and contribute to sustaining aberrant SGEC activation, which supports disease maintenance and perpetuates tissue inflammation. Such a detrimental role of autophagy is further indicated by evidence suggesting a role in determining the redistribution of autoantigens (mainly Ro-SSA) on the cell membrane following endoplasmic reticulum stress [38,41]. Taken together, these data raise the possibility of a beneficial effect of targeting autophagy in SS, as preliminarily demonstrated in mice models [42,43,44,45].

## 5. Autophagy Pathway Modulating Therapies

Many drugs currently used for the treatment of rheumatic diseases, such as glucocorticoids, hydroxychloroquine, rapamycin, anti-TNFα, and Jak inhibitors, can modulate different levels of autophagy.

Glucocorticoids seem to induce autophagy by inhibiting mTOR phosphorylation [45], and a proautophagic effect, sustaining proliferation and protection from apoptosis, has been demonstrated in bone marrow mesenchymal stem cells [46]. Besides glucocorticoids, hydroxychloroquine is another well-known therapy able to prevent autophagosome–lysosome fusion via interferences with lysosomal acidification [47]. This property allows considering hydroxychloroquine as one of the major drugs used in autoimmune rheumatic conditions modulating autophagy.

In 2018, a phase 1/2 trial evaluated the possible benefit of a known autophagy inhibitor, sirolimus (rapamycin), in SLE patients with clinically active disease resistant to, or intolerant of, conventional medications [47]. Results from this study demonstrated progressive suppression of disease activity during 12 months of sirolimus administration, suggesting a potential clinical benefit of this treatment in patients with SLE [48].

In 2019, we also demonstrated that belimumab, an inhibitor of B-lymphocyte stimulator (BLyS), is able to inhibit the apoptotic effect induced by BLyS on the progenitor of SLE endothelial cells, thus providing the first evidence of the modulatory property of this treatment on endothelial cell autophagy [49]. More recently, we also tested this autophagic modulatory property on PBMCs from patients with SLE treated with belimumab [50]. After 12 weeks of treatment with belimumab, a decrease in SLE PBMC autophagy, demonstrated by a significant reduction in LC3-II levels, was observed [50]. Taken together, these findings suggest how modulation of autophagic mechanisms in different cell types represents a potential additional mechanism by which belimumab downregulates disease activity in patients with SLE.

TNF-alpha inhibitors, whose prescription is largely diffused in patients with RA, seem also able to modulate the autophagy pathway. Indeed, it is well known that TNFα can induce autophagy in different cell types directly associated with RA pathogenesis and that TNFα-mediated autophagy may play a role in apoptosis resistance [51]. Indeed, Wang and colleagues showed that TNF-α can activate autophagy in FLS and that autophagy block may upregulate apoptosis and inhibit cell proliferation [51]. Thus, the block of autophagy by anti-TNF drugs in FLS may help in restoring this balance. Other evidence supports this hypothesis, demonstrating activation of synovial apoptosis after 8 weeks of treatment with two different classes of TNF-alpha inhibitors (etanercept and infliximab) [52]. In our previous work, Vomero et al. provided evidence of direct involvement of autophagy in the response to therapy in patients with RA [53]. Specifically, we analyzed changes in spontaneous autophagy in peripheral cells from patients with RA treated with TNF inhibitors, demonstrating a reduction in autophagy only in patients responding to the therapy [53]. This finding leads to the hypothesis that the level of autophagy can not only be considered a measure of disease activity but can also be looked at as a measure to identify patients more likely to respond to TNFα inhibitors.

Increasing evidence in the literature also demonstrates a link between the autophagy pathway and the Jak-STAT pathway [54,55,56,57]. Janus kinases (JAKs) are a family of nonreceptor tyrosine kinases, composed of four members, JAK1, JAK2, JAK3, and TYK2. JAKs are involved in different inflammatory and autoimmune diseases, and tofacitinib and baricitinib are selective JAK inhibitors that preferentially inhibit JAK1 and JAK3 and JAK1 and JAK2, respectively. JAK inhibitors are indicated for the treatment of moderate to severe RA patients who respond inadequately to at least one DMARDs. As mentioned above, data from other studies seem to suggest a role for Jak-STAT inhibition in autophagy modulation. In this regard, in RA, very recent in vitro studies evaluated the connection between the regulation of homeostatic mechanisms and the Jak-STAT pathway [58]. In SS, the first evidence of a potential utility of the use of baricitinib has been speculated thanks to the additional capacity to interfere with the autophagy pathway; however, larger studies are needed to clarify this aspect [59].

## 6. Conclusions

Autophagy is a homeostatic physiological process able to participate in the immunopathogenesis of different autoimmune disorders. The autophagy pathway increases the survival of immune and nonimmune cells, regulating peptide citrullination, the presentation of autoantigens, and promoting B- and T-cell maturation. Evidence of maladaptive activation of autophagy in different autoimmune rheumatic conditions supports the potential utility of drugs targeting this pathogenic pathway. Of note, some currently used treatments for different rheumatic diseases are capable of modulating cell homeostatic function, making autophagy manipulation an additional mechanism by which they achieve a beneficial therapeutic function. Further studies are warranted to clarify the exact mechanisms underlying autophagy pathogenic function in autoimmune rheumatic diseases and to evaluate the efficacy, safety, and long-term effects of autophagy modulation in these conditions.

## Figures and Tables

**Figure 1 cells-11-01359-f001:**
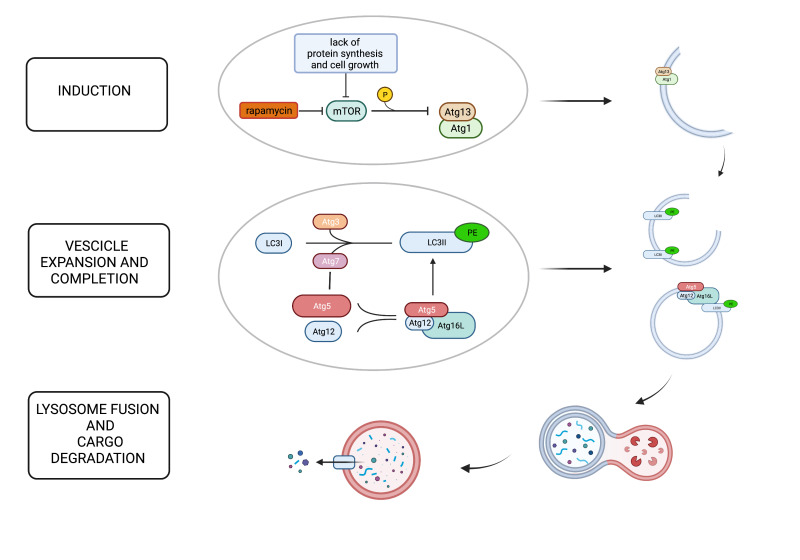
The figure shows the principal steps of the autophagy process: induction, elongation and completion, lysosome fusion, and cargo degradation. Mammalian target of rapamycin (mTOR) plays a pivotal role in the inhibition of autophagy phosphorylating Atg13 and preventing its association with Atg1, which is critical for autophagosome formation. When cell growth and protein synthesis are scarce, mTOR is inactivated, and autophagy is induced as a compensatory mechanism. Rapamycin is also a potent mTOR inhibitor. Autophagosome formation requires two ubiquitin-like systems: the ATG12 conjugation system and the ATG8 (LC3) conjugation system. Atg5 is covalently conjugated to Atg12, facilitated by Atg7. The Atg12–Atg5 dimer then binds Atg16L, forming a multiprotein complex, which is recruited to the forming autophagosome’s isolation membrane through Atg16L. In the second conjugation system, Atg7 and Atg3 mediate the conjugation of LC3 to the lipid phosphatidylethanolamine (PE). LC3II, the membrane-bound, autophagosome-associated form, is required for membrane expansion. The last step is autophagosome–lysosome fusion to generate autolysosomes that degrade the cargo.

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
