# Peer review of "Autophagy in Rheumatic Diseases: Role in the Pathogenesis and Therapeutic Approaches"

_cells, 2022, doi:10.3390/cells11081359_

Round 1

Reviewer 1 Report

This is a concise and excellent review of the role of autophagy in the pathogenesis of rheumatoid arthritis, systemic lupus erythematosus, and Sjogren syndrome. I have few minor comments below.

  1. The title may be modified to include not only RD but aslo SLE and SS.
  2. The following sentence in the introduction "Atg5 is another protein critical for the elongation of the autophagosome membrane during vacuole formation.": Atg5 is critical for completion of autophagome formation and subsequent degradation, not elongation (PMID: 27885029).

Author Response

Response to Reviewer #1 comments

Autophagy in rheumatic diseases: role in the pathogenesis and therapeutic approaches

This is a concise and excellent review of the role of autophagy in the pathogenesis of rheumatoid arthritis, systemic lupus erythematosus, and Sjogren syndrome. I have few minor comments below.

We thank the Reviewer for the appreciation of our study.

  1. The title may be modified to include not only RD but also SLE and SS.

Response 1: We decided to use the term “Rheumatic” to include RA, as well as SLE and SS, that are all considered rheumatic diseases.

  1. The following sentence in the introduction "Atg5 is another protein critical for the elongation of the autophagosome membrane during vacuole formation.": Atg5 is critical for completion of autophagome formation and subsequent degradation, not elongation (PMID: 27885029).

Response 2:  We agree with the reviewer that Atg5 is a crucial protein implicated in the completion of autophagosome formation. Accordingly, the above mentioned sentence at page 2 has been modified as follows: “Atg5 is a protein critical for the for completion of autophagosome formation”.

Reviewer 2 Report

In this review, the authors have tried to reveal how the crosstalk between autophagy and apoptosis can be dysregulated in autoimmune diseases with evidence of an aberrant expression of autophagic process that contributes to the pathogenesis of different rheumatic conditions including rheumatoid arthritis (RA), systemic lupus erythematosus (SLE), and Sjögren syndrome (SS). In addition to, they have proposed current and future therapeutic strategies for these autoimmune conditions. They have also reported that will be considered particular attention to their capacity of targeting autophagy as part of their mechanism of action in this review.

This is a well-designed and written review. However, in this review, which talks about autophagy and apoptosis, an important factor that must be present is not given.

This important shortcoming is the absence of figures in the review. However, such a subject had to be evaluated with figures.

My suggestions are as follows;

  1. Draw a figure with your own hands about autophagy and apoptosis that you have presented in the text of the introduction. This will make it easier for the reader to understand. Do not forget that this figure will make it easier for clinicians dealing with the three rheumatic diseases (RA, SLE and Sjogren) you have pointed out.
  2. Redraw similar shapes for each disease (RA, SLE and Sjogren, respectively) and place them under the text section of each.

Thus, your article will be quite understandable for the readership, particularly clinicians.

I would like to re-evaluate the article with additional it's figures, if it is re-written.

Reviewer 3 Report

The manuscript of Alessandra Ida Celia et al. is dedicated to a literature review on recent data on involvement of autophagy in pathogenesis of autoimmune rheumatic diseases. This field of investigation is significant because rheumatic diseases is surely an important biomedical problem and clarification of poorly understood mechanisms of the diseases is of importance for development of novel effective therapeutic approaches to treat these diseases. The significance of this review is beyond doubt, since the presented summary of literature data and their analysis give an objective integral picture of existing ideas on the indicated topic and highlight possible ways of practical use of the accumulated knowledge. These include information concerning the role of imbalance in the processes of autophagy and apoptosis in different cell populations for pathogenesis of autoimmune rheumatic diseases. An important aspect of the work is the discussion of the possibility of using autophagy inhibitors to treat a number of rheumatic diseases.

In general the review is well presented; the data are of considerable novelty and interest.

Several minor suggestions might improve the overall quality of the manuscript.

  1. Page 1. “…heat shock 71 protein (HSPA8) …” should be corrected as “…heat shock protein 70 (HSPA8) …” according to modern HSP classification.
  2. Page 2. “…with a potent autophagy inducer, rapamycin, human…”. “Inducer” should be replaced by “inhibitor”.
  3. Page 4. “ can also activate IKK which in turns modulates…”. IKK abbreviation should be deciphered.
  4. Page 5. “…demonstrated a progressive improvement in disease activity during 12 months…”. Substitution of “improvement” by “suppression” will be more correct.

Author Response

Response to Reviewer #3 comments

Autophagy in rheumatic diseases: role in the pathogenesis and therapeutic approaches

The manuscript of Alessandra Ida Celia et al. is dedicated to a literature review on recent data on involvement of autophagy in pathogenesis of autoimmune rheumatic diseases. This field of investigation is significant because rheumatic diseases is surely an important biomedical problem and clarification of poorly understood mechanisms of the diseases is of importance for development of novel effective therapeutic approaches to treat these diseases. The significance of this review is beyond doubt, since the presented summary of literature data and their analysis give an objective integral picture of existing ideas on the indicated topic and highlight possible ways of practical use of the accumulated knowledge. These include information concerning the role of imbalance in the processes of autophagy and apoptosis in different cell populations for pathogenesis of autoimmune rheumatic diseases. An important aspect of the work is the discussion of the possibility of using autophagy inhibitors to treat a number of rheumatic diseases.

In general the review is well presented; the data are of considerable novelty and interest.

We thank the Reviewer for the positive comments.

Several minor suggestions might improve the overall quality of the manuscript.

  1. Page 1. “…heat shock 71 protein (HSPA8) …” should be corrected as “…heat shock protein 70 (HSPA8) …” according to modern HSP classification.

Response 1: According to the modern HSP classification, the “heat shock 71 protein” has been modified in “heat shock 70 protein”.

  1. Page 2. “…with a potent autophagy inducer, rapamycin, human…”. “Inducer” should be replaced by “inhibitor”.

Response 2: We thank the reviewer for the comment. However, we think that inhibitor would change the meaning of the sentence. Rapamycin is in fact a potent inducer of autophagy in a diverse range of cell lines that works inhibiting mTOR action. (doi:10.1111/bcp.12893)

  1. Page 4. “can also activate IKK which in turns modulates…”. IKK abbreviation should be deciphered.

Response 3: IKK has been modified in “inhibitor of NF-κB kinase”

  1. Page 5. “…demonstrated a progressive improvement in disease activity during 12 months…”. Substitution of “improvement” by “suppression” will be more correct.

Response 4: The sentence at page 5 has been modified in “progressive suppression of disease activity”.

Reviewer 4 Report

The submitted manuscript is a concise review on the assumed role of autophagy in connective tissue disease (RA, SLE, SS). Overall, the review is of interest and inspiring. My most important concern is the unclear concept underlying the outlined interplay between autophagy and apoptosis.

Major:

  1. As outlined, increased autophagy leads to a decrease in apoptosis. While this concept gets relatively clear with regard to mechanisms in RA (in short, increased autophagy leads to a decrease in apoptosis of synoviocytes and increase in citrullination, leading to inflammation and autoantibody formation), the data seem to be somewhat contradictory in SLE (there is an apparent contradiction between decreased phagocytosis of apoptotic material and increased autophagy leading to the survival of phagocytes). In my understanding, there are data supporting a role for autophagy in SLE (supported by a number of observations), but a clear cut concept is missing. If this view is correct, the chapter on SLE should more clearly state this lack of understanding (but keep the message that a lot of data point to a role of autophagy), and/or maybe provide a figure how the authors interpret the observations made thus far. Similarly: If steroids induce autophagy, how can they reduce inflammation in RA at the same time. Mechanisms are likely to be more complex, but in its current format, the manuscript leads to some confusion.
  2. In SS, data from patients and mouse models seem to be contradictory (second paragraph of the chapter on SS). Again, please outline controversies more clearly in case there is (not yet) a common hypothesis.

Minor:

In the introduction, the paragraph on Atg5 jumps to other Atgs with a “Thus” (seems not to be logical) and back to Atg5. Maybe rewrite this paragraph or chose different wording.

Please stay consistent (beclin1 versus Beclin1, and Atg7 versus ATG7 when talking about a molecule (see chapter on RA))

There are a number of spelling errors (plural versus singular etc.) which should be easy to avoid.

Author Response

Response to Reviewer #4 comments

Autophagy in rheumatic diseases: role in the pathogenesis and therapeutic approaches

The submitted manuscript is a concise review on the assumed role of autophagy in connective tissue disease (RA, SLE, SS). Overall, the review is of interest and inspiring. My most important concern is the unclear concept underlying the outlined interplay between autophagy and apoptosis.

Major:

  1. As outlined, increased autophagy leads to a decrease in apoptosis. While this concept gets relatively clear with regard to mechanisms in RA (in short, increased autophagy leads to a decrease in apoptosis of synoviocytes and increase in citrullination, leading to inflammation and autoantibody formation), the data seem to be somewhat contradictory in SLE (there is an apparent contradiction between decreased phagocytosis of apoptotic material and increased autophagy leading to the survival of phagocytes). In my understanding, there are data supporting a role for autophagy in SLE (supported by a number of observations), but a clear cut concept is missing. If this view is correct, the chapter on SLE should more clearly state this lack of understanding (but keep the message that a lot of data point to a role of autophagy), and/or maybe provide a figure how the authors interpret the observations made thus far. Similarly: If steroids induce autophagy, how can they reduce inflammation in RA at the same time. Mechanisms are likely to be more complex, but in its current format, the manuscript leads to some confusion.

Response 1: We thank the Reviewer for the suggestions. We agree that many results regarding autophagy involvement in the pathogenesis of SLE are still controversial and in some cases, might be contradictory. We have now tried to better explain the possible link between autophagy dysregulation in monocyte and macrophages and the SLE pathogenesis. However, precise molecular mechanisms or processes have not been reported. We amplified the chapter at page 4, line 1-14 as follows “Autophagy is reported to be crucial for monocyte differentiation and for the prevention of regular apoptosis and survival of monocytes: inhibition of induced autophagy leads to apoptosis [24]. Therefore, autophagy shows abnormalities in lupus macrophages and autophagy-related genes are found to be up-regulated in macrophages of lupus mice and SLE patients, suggesting that autophagy may contribute to SLE pathogenesis via influencing monocytes and macrophages [25]. Macrophages from patients with SLE exhibit increased levels of autophagy, whereas in mice with a lupus­-like disease, inhibition of macrophage-­induced autophagy leads to decreased B­ cell maturation and reduced production of dsDNA [25]. Moreover, adoptive transfer of Beclin1 knockdown macrophages can significantly decrease anti- dsDNA antibody levels and immune complex deposition thus mitigating proteinuria and glomerulonephritis [25]. This protective effect seems to be associated with a significantly decreased production of IL-6 and TNF-α, indicating that abnormal activated autophagy in macrophages may contribute to lupus by promoting production of TNF-α and IL-6 [25].

Regarding steroids, we agree with Reviewer that mechanisms are likely to be more complex and our purpose was to mention the potential autophagy modulating drugs of clinical relevance in SLE. We have also modified the sentences at page 5 as follows: “Many drugs currently used for the treatment of rheumatic diseases, such as glucocorticoids, hydroxychloroquine, rapamycin, anti-TNFα and Jak inhibitors can modulate at different levels autophagy.

In SS, data from patients and mouse models seem to be contradictory (second paragraph of the chapter on SS). Again, please outline controversies more clearly in case there is (not yet) a common hypothesis.

Response 2: We thank the reviewer for this comment, in order to clarify this aspect the paragraph has been extensively implemented. See page 5 line 1.

Minor:

In the introduction, the paragraph on Atg5 jumps to other Atgs with a “Thus” (seems not to be logical) and back to Atg5. Maybe rewrite this paragraph or chose different wording.

Response 3: We understand reviewer’s comment. As “thus” might be confusing we decided to rephrase the sentence to clarify that the Atg3 and Atg7 mediated conjugation with Atg8 is a process driven by Atg5. Additionally, to avoid the idea that Atg5 related discussion is repeated twice in two different sections, we linked the concepts as follows. See page 2 , Line 9 “Atg5 is a protein critical for the for completion of the autophagosome formation, although a role in apoptotic processes has been described too. Regarding its role in autophagy activation, after conjugation with Atg12, Atg5 binds to Atg16L and forms a large multi-protein complex (first conjugation system) eventually recruited to the forming autophagosome’s isolation membrane. In the second conjugation system, Atg7 and Atg3 mediate the conjugation of Atg8 (LC3) to the lipid phosphatidylethanolamine resulting in the conversion from its soluble cytoplasmic form (LC3-I) to the membrane-bound autophagosome-associated form (LC3-II), which is required for membrane expansion [6] Figure 1.

However, as above mentioned, it is interesting to note that Atg5 also has a role in the apoptotic signaling. Specifically, after cleavage, Atg5 forms an N-terminal product that translocate to the mitochondria and promotes cytochrome C release and caspase activation [6]. Thanks to this capacity, Atg5 is considered a link between the autophagy and the apoptotic pathway [6]. ”

Please stay consistent (beclin1 versus Beclin1, and Atg7 versus ATG7 when talking about a molecule (see chapter on RA))

There are a number of spelling errors (plural versus singular etc.) which should be easy to avoid.

We thank the Reviewer, the manuscript has been revised accordingly.

Round 2

Reviewer 2 Report

In this way, the manuscript is much more useful for the readers.

Author Response

We thank the Reviewer for helping us to improve the manuscript. 

Reviewer 4 Report

The manuscript is surely improved but I still have comments that should be addressed:

  1. I cannot find the graphical abstract in the clean version of the manuscript (only in the response to reviewer 2). Why not placing it as Figure 2 in the manuscript ?
  2. The new paragraph on autophagy in SLE starting with “Autophagy is reported to be crucial …” needs some small corrections as the wording appears not to be ideal/partially misleading. The start of the second sentence with ‘therefore’ seems not to be logical. Maybe just delete the word or replace by “Interestingly” or “Of note”. In the same sentence: The word “contribute” does not seem to well cover what is happening. Better use “may be involved” or “…suggesting that increased autophagy may contribute …”. In the following sentence the use of ‘whereas’ seems to point to an opposite effect in mice, but this is not the case. Thus, maybe better use just ‘and’.

Author Response

Response to Reviewer #4 comments

Autophagy in rheumatic diseases: role in the pathogenesis and therapeutic approaches

The manuscript is surely improved but I still have comments that should be addressed:

  1. I cannot find the graphical abstract in the clean version of the manuscript (only in the response to reviewer 2). Why not placing it as Figure 2 in the manuscript ?
  2. The new paragraph on autophagy in SLE starting with “Autophagy is reported to be crucial …” needs some small corrections as the wording appears not to be ideal/partially misleading. The start of the second sentence with ‘therefore’ seems not to be logical. Maybe just delete the word or replace by “Interestingly” or “Of note”. In the same sentence: The word “contribute” does not seem to well cover what is happening. Better use “may be involved” or “…suggesting that increased autophagy may contribute …”. In the following sentence the use of ‘whereas’ seems to point to an opposite effect in mice, but this is not the case. Thus, maybe better use just ‘and’.

Response : We thank the Reviewer for the suggestions.

  1. We included the graphical abstract in the manuscript as Figure 2
  2. We replaced the term “therefore” with “of note” at page 4 line 1; the term “contribute” with “may be involved in” at page 4 line 5 and the term “whereas” with “and” at page 4 line 7.